# InfiniteAudio: Infinite-Length Audio Generation with Consistent Acoustic Attributes

## Abstract

This work aims to generate long-duration audio while preserving acoustic coherence, utilizing existing text-conditional audio generation models through diffusion-based approaches. Current diffusion models, however, encounter significant challenges in generating long audio sequences due to memory constraints, as output size scales with input length. While one possible solution is to concatenate short clips, this often leads to inconsistencies due to a lack of shared temporal information across segments. To address these challenges, we propose InfiniteAudio, a novel inference technique designed to generate long audio with consistent acoustic attributes. Our method is based on three key components. First, we implement a curved denoising approach with a fixed-size input, enabling theoretically infinite audio generation while maintaining a constant memory footprint. Second, we introduce conditional guidance alternation, a mechanism that enhances intelligibility in long speech generation. Finally, initial self-attention features are shared across future frames to maintain temporal coherence. The effectiveness of InfiniteAudio is demonstrated through comprehensive comparisons with existing text-to-audio generation baselines. Generated audio samples are available on our anonymous project page[1].

## 1 Introduction

Diffusion models (Ho et al., 2020; Song et al., 2020b) have received considerable attention across various domains due to their ability to generate high-quality, diverse outputs. They have demonstrated impressive results in tasks including image generation (Dhariwal & Nichol, 2021; Rombach et al., 2022), video generation (Ho et al., 2022; Singer et al., 2022; Wang et al., 2023), and text-to-audio (TTA) generation (Liu et al., 2023; Huang et al., 2023; Lee et al., 2024; Liu et al., 2024). TTA models generate audio from text description prompts and typically utilize generative frameworks such as latent diffusion models (Rombach et al., 2022) as illustrated in Fig. 1(a) or flow matching models (Vyas et al., 2023). Recently, VoiceLDM (Lee et al., 2024) has advanced this capability by generating both speech and background audio simultaneously, as shown in Fig. 1(b). The generated speech not only reflects the background description prompt but also adapts to the content prompt. For example, when prompted with "Hello" in a cathedral setting, the speech will naturally include reverberation to match the environment.

Despite these advancements, existing TTA generation models based on diffusion approaches face significant challenges when generating longer audio sequences. To extend the output size during inference, the input size must also be increased, given that diffusion models require the input and output dimensions to remain unchanged. Moreover, these models struggle to manage long text conditions when producing extended speech. While long audio can be generated by concatenating short clips created by existing TTA models, ensuring a smooth and continuous audio stream remains challenging due to the lack of temporal consistency between inter-clip segments.

To address these challenges, we introduce InfiniteAudio, a novel inference method for generating long and consistent audio. InfiniteAudio generates extended audio by utilizing a fixed input size with progressively increasing noise levels over time. As shown in Fig. 2, at each inference step, the fully denoised audio segment at the beginning of the input is removed, while a new random noise

---

[1]https://anonymousforcf.github.io/InfiniteAudio/

Figure 1: Overview of tasks. (a) InfiniteAudio enables the generation of longer audio using a pre-trained text-to-audio model, overcoming the memory limitations faced by existing models. (b) For simultaneous audio and speech generation, InfiniteAudio can generate long speech that accurately reflects the audio description prompt.

Table 1: Comparison of existing diffusion inference methods with our approach. Our method generates longer audio with a fixed memory size.

| Methods | Memory requirements | Long generation | Varying timesteps |
|---|---|---|---|
| Diffusion | Various | Limited | ✗ |
| FIFO-Diffusion (Kim et al., 2024) | Small | capable | ✓ |
| InfiniteAudio | Very small | capable | ✓ |

latent is added at the end. In this manner, InfiniteAudio can theoretically generate infinite audio frames using a fixed input size, effectively mitigating memory constraints.

While FIFO-Diffusion (Kim et al., 2024), which is designed for text-to-video (TTV) generation, also employs a fixed input size, it utilizes all diffusion sampling steps. In contrast, as illustrated in Fig. 2, our method chooses the more important steps rather than using the entire steps. This selective approach, which we refer to as curved denoising, reduces the number of required sampling steps while attaining high-quality generation, resulting in more efficient inference. Tab. 1 presents a comparison of traditional diffusion inference methods, FIFO-Diffusion, and our proposed approach.

Additionally, InfiniteAudio addresses the challenge of generating long audio sequences from extended text inputs by segmenting the text and applying a guidance alternation technique. By dividing long text prompts into smaller segments, we reduce memory overhead. However, when processing consecutive prompts, the generated audio can be affected by preceding segments, leading to reduced intelligibility. To resolve this, we propose a guidance alternation strategy that switches between conditional and unconditional guidance when processing following text inputs. This approach preserves the clarity of long speech while minimizing interference between segments.

While this method effectively handles extended speech, generating audio from multiple distinct prompts within a single clip can disrupt coherence, as the prompts often lack the specificity required to retain consistent speaker characteristics. To mitigate this issue, we share a query, key, and value (QKV) features within the self-attention layers of the diffusion model. Propagating the initial QKV features across successive segments ensures uniform speaker attributes and continuity, preserving vocal consistency and maintaining intelligibility of both background audio and speech, even with varying text inputs. Our experiments demonstrate that InfiniteAudio can generate extremely long and coherent audio without any degradation in quality over time. Further details are provided in Sec 4.

Our contributions can be summarized as follows.

- We propose InfiniteAudio, a method for generating long-duration audio without additional training, addressing memory limitations in existing TTA models using diffusion techniques.

- We introduce curved denoising, which selectively applies key diffusion steps, improving efficiency.

- We suggest a conditional guidance alternation mechanism to support multiple speech conditions within a single audio stream, maintaining intelligibility.

- We implement QKV sharing in self-attention, ensuring consistent speech generation.

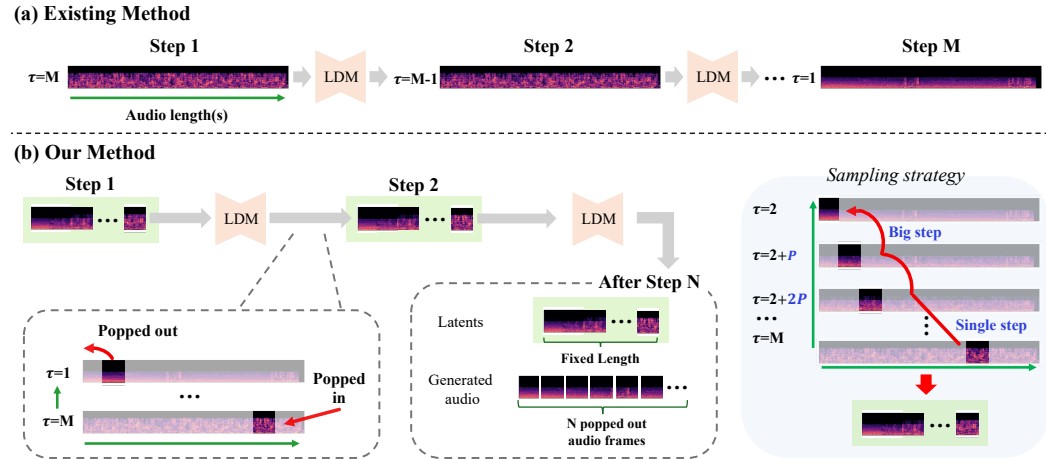

Figure 2: Overall pipeline for the existing method and our method. Traditional diffusion models apply the same diffusion timestep across inputs during inference. Our method starts with a latent containing varying timesteps and skips unimportant timesteps for $P$ multiple big steps. For every inference step, an audio frame reaching $\tau = 1$ is popped out and an audio frame with noise is inserted to maintain a same input size. This method theoretically allows infinite audio generation with constant memory usage, producing one audio frame per step.

## 2 RELATED WORKS

### 2.1 TEXT TO AUDIO AND SPEECH GENERATION

TTA generation (Liu et al., 2023; Kreuk et al., 2022; Yang et al., 2023) has attracted considerable attention in recent years, driven by advancements in generative modeling techniques (Ho et al., 2020; Song et al., 2020b). Several works (Liu et al., 2023; Ghosal et al., 2023; Yang et al., 2023) use the latent diffusion model (LDM) (Rombach et al., 2022) to generate audio, mitigating the large computational costs of the original diffusion process. In the diffusion-based TTA models, contrastive language audio pretraining (CLAP) (Wu et al., 2023) is utilized in many models (Liu et al., 2023; Huang et al., 2023; Yuan et al., 2024), in order to align language and audio embeddings. Additionally, large language models (LLMs) are exploited due to their strong text understanding capabilities (Ghosal et al., 2023; Liu et al., 2024).

Besides TTA, text-to-speech (TTS) generation is also an active area of research, with early models using autoregressive (AR) models (Wang et al., 2017; Oord et al., 2016). To address the issue of slow inference speed that arises in AR models, researchers have introduced non-AR models (Ren et al., 2019; Kim et al., 2020) that offer improved performance compared to AR models. Using the diffusion model, Grad-TTS (Popov et al., 2021) produces high-quality speech with a score-based decoder. Furthermore, several works have addressed environment-related speech generation. For example, VoiceLDM (Lee et al., 2024) introduces an efficient model that generates audio closely aligned with both descriptive and content prompts. Audiobox (Vyas et al., 2023), a unified model based on flow matching, can produce audio that contains various audio conditions, such as non-verbal sounds (e.g., coughing, screaming) or acoustic environments (e.g., rural, stadium, indoor). While these environment-related speech generation models produce high-quality results, they struggle to generate long outputs that containing multiple sentences simultaneously.

### 2.2 LONG GENERATION USING DIFFUSION MODELS

Producing large-scale output with diffusion models is challenging due to the high computational costs and memory footprints. For image generation, Multi-Diffusion (Bar-Tal et al., 2023) and SyncDiffusion (Lee et al., 2023) use several windows to generate images with arbitrary aspect ratios but focus on smoothing the overlap regions of windows, falling short of solving the repetition problem. Scalecrafter (He et al., 2023) dynamically increases the receptive field and succeeds in generating ultra-high-resolution images up to $4096 \times 4096$.

In addition to image generation, research into long video generation has become increasingly active. Many AR models (He et al., 2022; Voleti et al., 2022; Harvey et al., 2022; Chen et al., 2023) can generate long videos, but due to error accumulation and a lack of temporal consistency between the frames, there are quality issues. FreeNoise (Qiu et al., 2023) addresses this issue with a window-based function but cannot generate infinitely long videos as it requires memory proportional to the output length. FIFO-Diffusion (Kim et al., 2024) can produce infinitely long videos with a fixed amount of memory by conducting diagonal diffusion across different timesteps.

In the audio domain, residual vector quantization (RVQ) is widely used to generate audio faster and more efficiently (Défossez et al., 2022; Zeghidour et al., 2021). SoundStorm (Borsos et al., 2023b), which combines RVQ with AudioLM (Borsos et al., 2023a), efficiently generates audio sequences up to 30 seconds long, a relatively extended length. To generate longer audio, (Evans et al., 2024a) tackles this issue by leveraging LDMs, resulting in output up to 95 seconds long. Moreover, (Evans et al., 2024b) leverage the diffusion transformer(DiT) to generate even longer audio, stretching up to 4m 45s. However, these approaches (Evans et al., 2024a;b) require additional training on datasets that match the desired output length. Our method does not require any additional training to generate audio of theoretically infinite length.

## 3 TEXT-TO-AUDIO DIFFUSION MODELS

We briefly summarize the outline of existing text-to-audio generation models, focusing on two representative models: AudioLDM (Liu et al., 2023) and VoiceLDM (Lee et al., 2024). TTA models produce audio based on given text prompts and deal with audio as an image since audio can be represented as a 2D mel-spectrogram consisting of time and frequency axes.

Many TTA models commonly include the following modules: audio and text encoders, an audio decoder and vocoder, and a latent diffusion model. Through these modules, TTA models can learn the distribution of mel-spectrograms corresponding to a text prompt $y$. For the audio $f_{audio}(\cdot)$ and text encoder $f_{text}(\cdot)$, many models (Liu et al., 2023; Lee et al., 2024; Huang et al., 2023) leverage a contrastive language-audio pretraining (CLAP) encoder, which is trained to align text and audio modalities (Wu et al., 2023). These encoders encode a start latent and conditions that are exploited in the latent space, while the decoder reconstructs the mel-spectrogram denoted by $a \in \mathbb{R}^{T \times F}$ from the latent $\mathbf{z}_1 \in \mathbb{R}^{C \times \frac{T}{r} \times \frac{F}{r}}$, where $T$ represents the time dimension, $F$ denotes the frequency dimension, $C$ refers to the channel dimension, $\tau \sim \mathcal{U}([1, ..., M])$ is the diffusion timestep, and r is the compression factor. The vocoder produces a waveform from the predicted mel-spectrogram. The LDM is trained to denoise a perturbed version of the latent from $\mathbf{z}_\tau$ to $\mathbf{z}_1$.

For noise $\epsilon \sim \mathcal{N}(\mathbf{0}, \mathbf{I})$ and the text condition $\mathbf{c} = f_{text}(y)$, AudioLDM is trained to minimize the following loss:

$$\mathcal{L}_{AudioLDM} = \mathbb{E}_{\mathbf{z}_0, \boldsymbol{\epsilon}, \tau} \left[ \|\boldsymbol{\epsilon} - \boldsymbol{\epsilon}_\theta(\mathbf{z}_\tau, \tau, \mathbf{c})\|_2^2 \right]. \tag{1}$$

, where $\epsilon_\theta$ is the predicted diffusion score. Unlike AudioLDM, VoiceLDM generates not only general audio but also produces clean speech and speech that reflects background sounds. Therefore, the model uses two text prompts: a description prompt $\mathbf{c}_{desc}$ and a content prompt $\mathbf{c}_{cont}$. Similar to AudioLDM, VoiceLDM uses a CLAP encoder and latent diffusion model architectures. The objective for VoiceLDM is as follows:

$$\mathcal{L}_{VoiceLDM} = \mathbb{E}_{\mathbf{z}_0, \boldsymbol{\epsilon}, \tau} \left[ \|\boldsymbol{\epsilon} - \boldsymbol{\epsilon}_\theta(\mathbf{z}_\tau, \tau, \mathbf{c}_{desc}, \mathbf{c}_{cont})\|_2^2 \right]. \tag{2}$$

The model uses dual classifier-free guidance (Ho & Salimans, 2022) to control audio description prompt and a text content prompt. Therefore, the diffusion score $\tilde{\epsilon}$ is formulated as follows:

$$\begin{aligned} \tilde{\epsilon}_\theta \left( \mathbf{z}_\tau, \mathbf{c}_{desc}, \mathbf{c}_{cont} \right) = {} & \boldsymbol{\epsilon}_\theta \left( \mathbf{z}_\tau, \mathbf{c}_{desc}, \mathbf{c}_{cont} \right) \\ & + w_{desc} \left( \boldsymbol{\epsilon}_\theta \left( \mathbf{z}_\tau, \mathbf{c}_{desc}, \emptyset_{cont} \right) - \boldsymbol{\epsilon}_\theta \left( \mathbf{z}_\tau, \emptyset_{desc}, \emptyset_{cont} \right) \right) \\ & + w_{cont} \left( \boldsymbol{\epsilon}_\theta \left( \mathbf{z}_\tau, \emptyset_{desc}, \mathbf{c}_{cont} \right) - \boldsymbol{\epsilon}_\theta \left( \mathbf{z}_\tau, \emptyset_{desc}, \emptyset_{cont} \right) \right) \end{aligned} \tag{3}$$

where $w$ is the guidance weight and $\emptyset$ indicates the null condition.

## 4 INFINITEAUDIO

In this section, we describe how to generate infinite-length audio with a fixed memory footprint using pretrained TTA models. Additionally, we introduce a method for generating longer, consis-

tent speech. We explore two representative TTA models, AudioLDM and VoiceLDM, and suggest suitable inference techniques respectively.

## 4.1 CURVED DENOISING WITH REDUCED SAMPLING STEPS

Because diffusion models are trained to estimate the noise in the input, both the input and output must have the same size. This inherent property increases memory demands, as generating longer audio requires longer inputs. For instance, on a single GPU with 12GB of memory, AudioLDM is limited to generating audio no longer than 22.5 seconds.

To overcome memory limitations, we propose InfiniteAudio, which operates with a fixed input size but can theoretically generate infinite-length audio. Inspired by FIFO-Diffusion (Kim et al., 2024), which addresses text-to-video generation, we initiate the diffusion inference process with a fixed-length audio segment using a small portion of the output predicted by existing TTA models. Although mel-spectrograms, which consist of time and frequency axes, can be treated as images, audio generation tasks must handle temporal information, similar to video generation. Therefore, we treat the input latent $\mathbf{z}_\tau \in \mathbb{R}^{C \times \frac{T}{r} \times \frac{F}{r}}$ as containing $\frac{T}{r}$ audio frames, analogous to video frames. Each encoded mel-spectrogram frame corresponds to $\mathbf{z}_1^i \in \mathbb{R}^{C \times 1 \times \frac{F}{r}}$, where $i \in [1, \frac{T}{r}]$.

For infinite audio generation, noise is progressively added to the input audio frames over time, except for the initial frames, which act as a 'buffer zone' with no added noise. Since no further training occurs in InfiniteAudio, using different diffusion timesteps during inference can produce a performance gap. The buffer zone mitigates this by applying the same timesteps as during training, helping to reduce the performance gap. Beyond the buffer zone, the latent frames gradually transition: the earlier frames are almost fully predicted, whereas the $\frac{T}{r}$-th frame is treated as Gaussian noise. The input to the inference stage consists of the initial buffer frames and $\frac{T}{r}$ frames with varying noise levels. As represented in Fig. 2 (b), after each inference step, the first frame following the buffer zone reaches diffusion timestep $\tau = 1$ and is then removed. To maintain a total of $\frac{T}{r}$ frames, we insert a new noisy frame at the $\frac{T}{r}$-th position. By repeating this process iteratively, we can generate $N$ frames in $N$ inference steps. To effectively tackle memory limitation issues, InfiniteAudio keeps the input size constant during inference, regardless of the desired output length. However, employing the full set of diffusion timesteps still necessitates long input sequences. To mitigate this issue, InfiniteAudio reduces input size by selecting only the most critical diffusion steps. By leveraging deterministic denoising (Song et al., 2020a), existing models perform inference without requiring all timesteps. Additionally, we found that we can further reduce the number of timesteps by skipping unimportant steps while still preserving sample quality.

We first identify the most important timesteps of the three segments, initial, middle, and final, during inference for both AudioLDM and VoiceLDM. Since the attention scores for both models reflect the relevance of one frame (Key) to another frame (Query), we analyze the self-attention maps in the diffusion U-Net decoder modules.

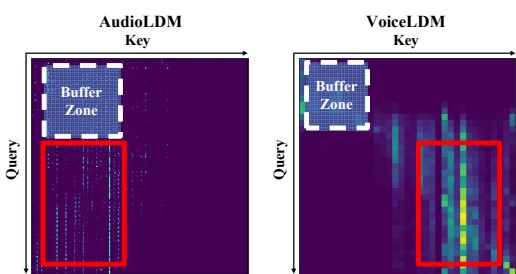

Figure 3: Attention maps denoting the importance of timesteps in the input sequences. In AudioLDM, the query in the last sequence segment focuses on the initial portions of the audio. In contrast, VoiceLDM demonstrates a stronger correlation with the later segments in its final query.

As shown in Fig. 3, in AudioLDM, the query sequences are primarily influenced by the initial key sequences, which correspond to earlier frames or cleaner inputs. In contrast, VoiceLDM behaves differently: the query sequences are more influenced by later key sequences, which correspond to noisier inputs. Since some initial frames lie within a buffer zone, we focus on regions beyond this zone.

Consequently, we allocate more timesteps to critical regions with high attention scores and skip less crucial timesteps, significantly reducing the overall number of inference steps and input size. This strategy, dubbed as curved denosing, enables us to achieve similar output quality with fewer computations compared to the traditional method, which uses $N$ timesteps for $N$ frames.

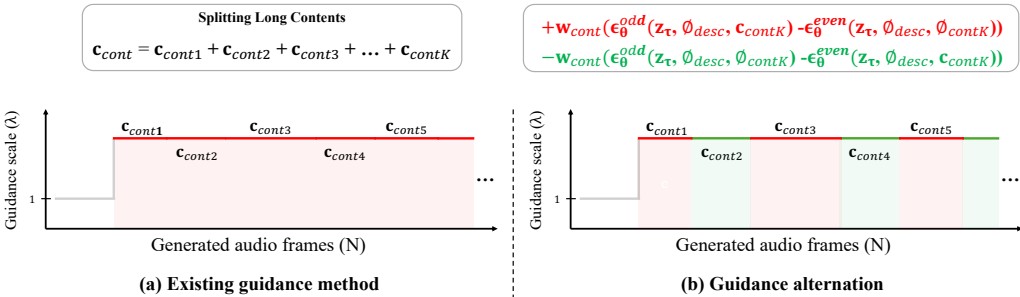

Figure 4: Illustration of guidance alternation method. Long sentence token $\mathbf{c}_{cont}$ is divided into several sentence tokens such as $\mathbf{c}_{cont1}$, $\mathbf{c}_{cont2}$, and so on. We apply existing conditional guidance to odd-numbered sentence prompts and switch to unconditional guidance for even-numbered sentence prompts. This alternation helps reduce the influence of one segment on the generation of the next, improving overall coherence in the generated audio.

## 4.2 LONG SPEECH GENERATION

In addition to generating long sounds, it is essential to generate extended speech. However, generating long speech with LDM by encoding a long content prompt at once with a text encoder is challenging due to memory limitations. To overcome memory limitation, we first split the long content prompt $\mathbf{c}_{cont}$ into smaller, manageable sentence segments: $\mathbf{c}_{cont1}, \mathbf{c}_{cont2}, ..., \mathbf{c}_{contk}$. Each segment is then applied to its corresponding audio section.

After generating the first sentence segment, the model faces challenges in immediately processing the next sentence prompt due to differing diffusion timesteps in the input. The part of latent which is in its final timesteps becomes confused when it receives a new sentence prompt, as it has already processed using the previous sentence prompt. Therefore, we need to eliminate the residual effects from the previous sentence before generating the next one. As shown in Fig. 4, rather than simply sequencing the sentence segments $\mathbf{c}_{cont2}, ..., \mathbf{c}_{contk}$, which causes interference between sentences, we devise a novel guidance alternation method.

Considering the guidance scale in Eq. 3, we utilize existing guidance differently depending on the sentence prompts are odd-numbered or even-numbered prompts denoted in Eq. 4 and 5 , respectively.

$$
\begin{aligned}
\tilde{\boldsymbol{\epsilon}}_\theta \left( \mathbf{z}_\tau, \mathbf{c}_{desc}, \mathbf{c}_{cont} \right) = {} & \boldsymbol{\epsilon}_\theta \left( \mathbf{z}_\tau, \mathbf{c}_{desc}, \mathbf{c}_{cont} \right) \\
& + w_{desc} \left( \boldsymbol{\epsilon}_\theta \left( \mathbf{z}_\tau, \mathbf{c}_{desc}, \emptyset_{cont} \right) - \boldsymbol{\epsilon}_\theta^{even} \left( \mathbf{z}_\tau, \emptyset_{desc}, \emptyset_{cont} \right) \right) \\
& + w_{cont} \left( \boldsymbol{\epsilon}_\theta^{odd} \left( \mathbf{z}_\tau, \emptyset_{desc}, \mathbf{c}_{cont} \right) - \boldsymbol{\epsilon}_\theta^{even} \left( \mathbf{z}_\tau, \emptyset_{desc}, \emptyset_{cont} \right) \right)
\end{aligned}
\tag{4}
$$

$$
\begin{aligned}
\tilde{\boldsymbol{\epsilon}}_\theta \left( \mathbf{z}_\tau, \mathbf{c}_{desc}, \mathbf{c}_{cont} \right) = {} & \boldsymbol{\epsilon}_\theta \left( \mathbf{z}_\tau, \mathbf{c}_{desc}, \mathbf{c}_{cont} \right) \\
& + w_{desc} \left( \boldsymbol{\epsilon}_\theta \left( \mathbf{z}_\tau, \mathbf{c}_{desc}, \emptyset_{cont} \right) - \boldsymbol{\epsilon}_\theta^{odd} \left( \mathbf{z}_\tau, \emptyset_{desc}, \emptyset_{cont} \right) \right) \\
& - w_{cont} \left( \boldsymbol{\epsilon}_\theta^{odd} \left( \mathbf{z}_\tau, \emptyset_{desc}, \emptyset_{cont} \right) - \boldsymbol{\epsilon}_\theta^{even} \left( \mathbf{z}_\tau, \emptyset_{desc}, \mathbf{c}_{cont} \right) \right)
\end{aligned}
\tag{5}
$$

For even-numbered sentence prompts, such as $\mathbf{c}_{cont2}, \mathbf{c}_{cont4}, \mathbf{c}_{cont6}$, we switch from conditional guidance to unconditional guidance to mitigate the influence of the previous sentence. Specifically, we treat existing unconditional guidance as conditional guidance and vice versa for odd-numbered prompts. The red-highlighted sections indicate the existing conditional guidance, while the green-highlighted sections represent the alternated conditional guidance. This approach helps ensure that sentences are generated accurately and completely. Furthermore, to balance the guidance alternation method, we apply a negative sign to the guidance weights, $w_{cont}$. As a result, we no longer need to input the entire sentence prompt, which significantly reduces memory and computational demands while enhancing speech intelligibility.

## 4.3 CONSISTENT SPEECH GENERATION

While long speech generation can be efficiently achieved by separating sentences and employing the guidance alternation method, it is essential to maintain consistent speaker attributes throughout the generation process. Splitting long sentences into multiple tokens can lead to inconsistencies,

such as fluctuations between male and female voices. To address this issue, we propose sharing query, key, and value (QKV) features within the self-attention layers of the U-Net architecture in the diffusion model, as demonstrated in Fig. 5. In video editing and image translation research, self-attention layers in the diffusion model are critical as they can determine the overall structure of the image (Tumanyan et al., 2023; Ceylan et al., 2023). Query, key, or value pairs of the previous image are used for the next image generation to maintain the overall image concept.

Unlike previous works that focus on maintaining consistency within a single image by utilizing a single image (Tumanyan et al., 2023; Ceylan et al., 2023), we aim for long-term consistency by sharing the QKV pairs from the initial speech throughout the generation of subsequent sentences in the U-Net upsampling layers. We retain a series of QKV features for the initial $T$ audio frames and utilize them during the generation of later sentences. We empirically demonstrate a method for selecting the appropriate QKV pairs and determining the optimal number of frames to share.

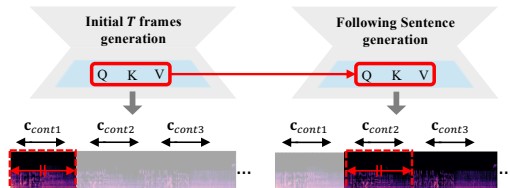

Figure 5: To maintain a speaker characteristic, QKV paris in initial $T$ audio frames are utilized. For following sentence generation, the model loads the QKV pairs for speech consistency.

## 5 EXPERIMENT

We present generated long audio using InfiniteAudio, built on pretrained AudioLDM and VoiceLDM models, and evaluate them quantitatively and qualitatively. Furthermore, we perform ablation studies on Sec. 4. For more audio samples and additional ablation studies, see App. B and D .

### 5.1 EXPERIMENTAL SETTINGS

**Datasets.** To evaluate our method on TTA generation, we randomly selected 500 audio-text pairs from the 975 test files in the Audiocaps dataset (Kim et al., 2019), which is commonly used for evaluating existing TTA models. For text-to-long-speech (TTLS) generation, we constructed a test set from the English subset of the CommonVoice 13.0 corpus (Ardila et al., 2019), randomly selecting 60 text samples, each consisting of more than five sentences, to assess long-form speech generation. For TTAS, which involves generating both audio and speech, we used 60 randomly selected text sets from the CommonVoice 13.0 corpus (Ardila et al., 2019). The test set for audio description was sourced from the Audiocaps test set (Kim et al., 2019), specifically focusing on samples from the "speech" category, such as those involving "talking" or "speaking."

**Baselines.** For comparison, we evaluate the performance of InfiniteAudio against two publicly available TTA models: AudioLDM[2] and VoiceLDM[3]. Notably, VoiceLDM is currently the only available model capable of generating both audio and speech simultaneously, making it the sole candidate for our TTLS and TTAS experiments.

**Evaluation Metrics.** We employ several quantitative metrics to evaluate the audio quality and the alignment between the input text prompt and the generated audio. These metrics include Frechet Distance (FD), Kullback-Leibler (KL) divergence, and the CLAP score, which are standard in text-to-audio generation evaluations (Liu et al., 2023; Lee et al., 2024; Vyas et al., 2023). FD and KL divergence quantify how closely the generated audio matches the ground truth, with lower values indicating better performance. The CLAP score, in contrast, assesses the relevance between the text prompts and the generated audio, where higher values are preferable. For subjective evaluation of the audio produced by TTA models, we use two metrics: (i) overall quality (OVL) and (ii) relevance to the input text description (REL). Both were rated on a scale of 1 to 5 by 20 domain experts, based on 30 speech samples. Further details on the human evaluation process are available in App. C.3.

---

Table 2: Quantitative evaluations on TTA. Our method for both models achieves comparable results, even surpassing original inference results.

| Method | CLAP↑ | FD↓ | KL↓ | OVL↑ | REL↑ |
|---|---|---|---|---|---|
| Ground Truth | 0.5276 | NA | NA | 4.11±0.22 | 4.03±0.25 |
| AudioLDM (Liu et al., 2023) | **0.4908** | 44.6689 | 2.0805 | **3.03±0.23** | **3.06±0.21** |
| InfiniteAudio w/ Equally spaced timesteps | 0.3832 | 54.7479 | 2.4013 | 2.19±0.21 | 2.33±0.23 |
| InfiniteAudio w/ Middle focused timesteps | 0.3979 | 56.7792 | 2.6077 | 2.06±0.19 | 2.18±0.20 |
| **InfiniteAudio w/ Last focused timesteps** | 0.4559 | **43.3788** | **1.9650** | 2.63±0.18 | 2.80±0.21 |
| InfiniteAudio w/ Initial focused timesteps | 0.3110 | 67.0704 | 2.9838 | 2.13±0.19 | 2.07±0.20 |
| VoiceLDM (Lee et al., 2024) | **0.4199** | 51.4019 | **2.2749** | **2.53±0.24** | 2.41±0.21 |
| InfiniteAudio w/ Equally spaced timesteps | 0.3729 | 59.1521 | 2.4477 | 2.20±0.21 | 2.33±0.22 |
| InfiniteAudio w/ Middle focused timesteps | 0.3779 | 56.7321 | 2.4622 | 2.10±0.20 | 2.41±0.22 |
| InfiniteAudio w/ Last focused timesteps | 0.3542 | 64.8813 | 2.6227 | 2.38±0.23 | 2.24±0.21 |
| **InfiniteAudio w/ Initial focused timesteps** | 0.4107 | 51.5047 | 2.3498 | 2.38±0.23 | **2.48±0.21** |

To assess speech intelligibility, we measure word error rate (WER) and character error rate (CER) using the Whisper automatic speech recognition (ASR) model (Radford et al., 2023), where lower scores indicate better intelligibility.

To evaluate voice consistency within a single audio sample, we utilize the Resemblyzer Python package[4], which is commonly employed for extracting speaker embeddings, alongside the VoxCeleb-disentangler model (Nam et al., 2024), which offers a high-level representation of speakers. We calculate the cosine similarity score $C_{sim}$ based on the first 10 seconds of the speaker embedding $G(a_{:10})$, where $G$ represents the speaker verification model used in the aforementioned methods. Additionally, we assess 5-second segments of the embedding, $G(a_{5+5h:10+5h})$, where $h = 1, 2, 3$, and $4$. This score $C_{sim}$ quantifies the similarity between two vectors in an inner product space.

## 5.2 QUANTITATIVE RESULTS

**Memory Consumptions.** We compare the memory consumption of the existing TTA models with our method. Since VoiceLDM restricts generation to 10 seconds, we conduct our experiments using AudioLDM. AudioLDM's memory usage increases as the length of the generated audio grows. In contrast, our method maintains consistent memory usage, regardless of the desired audio length, as demonstrated in Fig. 6. Consequently, our method allows for generating longer audio content without significant performance degradation.

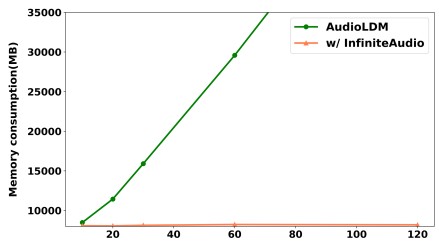

Figure 6: Comparisons on memory consumption between AudioLDM (Liu et al., 2023) and our method.

**Generation Evaluations.** We demonstrate the effectiveness of curved denoising strategy in Tab. 2. While our framework is designed to generate long audio with fixed memory based on a pre-trained model, it not only matches the performance of existing models but also achieves higher scores. We demonstrate that our approach considering input attention relations represented in Fig. 3 achieves superior performance compared to the equally spaced timesteps that are used in FIFO-Diffusion (Kim et al., 2024) or other strategies with the same steps.

Since VoiceLDM generates long speech in 10-second segments, it exhibits a significantly higher WER than InfiniteAudio, as demonstrated in Tab. 3. In contrast, InfiniteAudio's alternating guidance strategy further reduces both WER and CER, enhancing sentence intelligibility by mitigating interference between sentences in text-to-long speech (TTLS) generation. For TTAS evaluation, our method delivers superior performance over existing approaches, particularly in WER and CER scores, with only a slight reduction in the CLAP score. As there is no ground truth for generating both audio and speech simultaneously, we focus our evaluation on WER, CER, and CLAP scores.

---

[4]https://github.com/resemble-ai/Resemblyzer

Table 3: Evaluation on TTS and TTAS. Guidance alternation and QKV sharing method can further decreases WER, which contribute to speech intelligibility.

| Task | Method | WER↓ | CER↓ | CLAP↑ |
|------|--------|------|------|-------|
| TTLS | VoiceLDM | 0.5363 | 0.4595 | NA |
| | w/ InfiniteAudio | 0.5810 | 0.5119 | |
| | w/ InfiniteAudio and Guidance alternation | 0.3376 | 0.2635 | |
| | **w/ InfiniteAudio, Guidance alternation and QKV sharing** | **0.3038** | **0.2368** | |
| TTAS | VoiceLDM | 0.8038 | 0.6070 | **0.1252** |
| | w/ InfiniteAudio | 0.4604 | 0.3492 | 0.0988 |
| | w/ InfiniteAudio and Guidance alternation | 0.3863 | **0.2825** | 0.1200 |
| | **w/ InfiniteAudio, Guidance alternation and QKV sharing** | **0.3824** | 0.2888 | 0.0877 |

Table 4: Speaker consistency evaluation across different time regions. $C_{sim}^h$ represents the cosine similarity score between the first 10 seconds of speech and the subsequent 5 seconds. With QKV sharing, the speaker embeddings remain consistent throughout the entire duration.

| Task | Method | Resemblyzer | | | | VoxCeleb-disentangler (Nam et al., 2024) | | | |
|------|--------|-------------|--|--|--|-------------------------------------------|--|--|--|
| | | $C_{sim}^1 \uparrow$ | $C_{sim}^2 \uparrow$ | $C_{sim}^3 \uparrow$ | $C_{sim}^4 \uparrow$ | $C_{sim}^1 \uparrow$ | $C_{sim}^2 \uparrow$ | $C_{sim}^3 \uparrow$ | $C_{sim}^4 \uparrow$ |
| TTLS | InfiniteAudio | 0.7810 | 0.7564 | 0.7479 | 0.8218 | 0.4802 | 0.4336 | 0.4524 | 0.5481 |
| | **+ QKV sharing** | **0.7900** | **0.7680** | **0.7658** | **0.8415** | **0.5310** | **0.5207** | **0.5400** | **0.6154** |
| TTAS | InfiniteAudio | 0.8101 | 0.8082 | 0.8019 | 0.8718 | 0.5016 | 0.4983 | 0.4997 | 0.6096 |
| | **+ QKV sharing** | **0.8406** | **0.8183** | **0.8254** | **0.8810** | **0.5699** | **0.5280** | **0.5463** | **0.6236** |

**Voice Consistency.** The test set comprises 60 long text samples, identical to those utilized in the TTLS and TTAS evaluations. As shown in Tab. 4, sharing QKV yields speaker embedding features that are more closely aligned across the entire speech segments while preserving speech intelligibility.

## 5.3 QUALITATIVE RESULTS

**Sampling Strategies.** We propose curved denoising, where the sampling strategy is determined by considering attention scores. As shown in Fig. 7, in contrast to other strategies, which show interruptions in the generated audio as observed in the spectrograms, our method using initial-focused timesteps ensures continuous audio generation, as evidenced by both the spectrogram and the CLAP score.

**Generation Quality.** Fig. 8 illustrates the mel-spectrograms of the generated audio along with their corresponding evaluation scores. InfiniteAudio excels in TTA generation, produc-

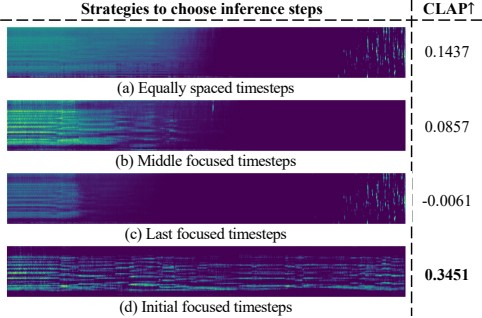

Figure 7: Analysis on various diffusion sampling strategies on VoiceLDM (Lee et al., 2024).

ing audio that adheres to the description prompts for durations exceeding 10 seconds. For evaluating TTLS generation, we utilize WER as the primary metric. Our method consistently produces intelligible speech, whereas VoiceLDM often struggles, frequently distorting speech segments to fit within a 10-second constraint. Generating coherent audio and speech simultaneously, especially for extended durations, is challenging due to the need to satisfy both the content prompt $\mathbf{c}_{cont}$ and the description prompt $\mathbf{c}_{desc}$. In contrast, InfiniteAudio effectively generates speech that aligns with both prompts.

## 5.4 ABLATION STUDY ON QKV SHARING

Figure 9 demonstrates our method for selecting QKV features, showing that sharing QKV features consistently outperforms other approaches across all metrics. Notably, it achieves higher cosine similarity scores, indicating better voice consistency, compared to the non-sharing method.

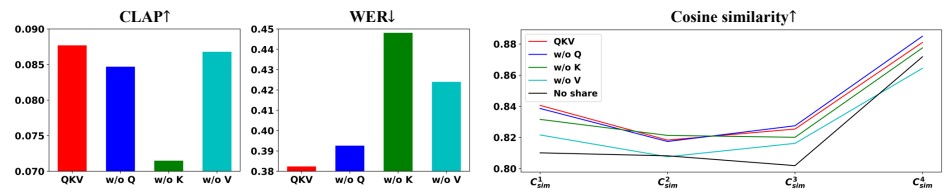

| Conditions | | Methods | | Score | |
|---|---|---|---|---|---|
| $c_{desc}$ | $c_{cont}$ | VoiceLDM | +InfiniteAudio | VoiceLDM | +InfiniteAudio |
| **Birds chirping** | ∅ | | ⋯ | CLAP↑ : 0.5247 | **CLAP : 0.5431** |
| **Clean Speech** | Harry looks up to the join works of the stairs with small amounts of dusts. Dudley comes down the stairs, and runs for the kitchen. Harry tries to come out of the closet but is pushed back in by Dudley. Harry leaves the living room area and picks up some letters, one of which had his name on it. | | ⋯ | WER↓ : 10.2% | **WER :1.7%** |
| **She is talking in a park.** | Dobby has no master, Dobby is a free elf! And Dobby has come to save Harry Potter and his friends. Every great wizard in history has started out as nothing more than we are now. Can you tell me where I might find platform Nine and Three-Quarters? | | ⋯ | CLAP : 0.3111 WER: 10.4% | **CLAP : 0.3581 WER : 6.2%** |

Figure 8: Qualitative results for TTA, TTLS, and TTAS. InfiniteAudio generates high-quality long audio that accurately follows both the audio description prompt and the speech content prompt.

Figure 9: Ablation study on QKV combinations for speaker consistency.

## 5.5 ANALYSIS ON SAMPLING STEPS AND AUDIO LENGTH

InfiniteAudio is designed to minimize the number of sampling steps while preserving high audio quality. As shown in Tab. 2, our method outperforms other strategies with the same number of steps. Furthermore, even when compared to methods that increase sampling steps to 200 or 250 using equally spaced timesteps, our approach consistently achieves excellent scores across all metrics, despite utilizing fewer than 150 steps, as denoted in Tab. 5.

Table 5: Comparison of sampling steps between VoiceLDM (Lee et al., 2024) and InfiniteAudio. InfiniteAudio requires fewer than 150 steps to achieve superior results.

| Sampling steps | CLAP↑ | FD↓ | KL↓ |
|---|---|---|---|
| w/ 200 equally spaced steps | 0.3923 | 53.0555 | **2.3334** |
| w/ 250 equally spaced steps | 0.3941 | **50.5447** | 2.3937 |
| **InfiniteAudio** | **0.4107** | 51.5047 | 2.3498 |

InfiniteAudio aims to generate longer audio sequences while maintaining high quality. As shown in Tab. 6, compared to the fixed 10-second generation, it produces comparable results across a range of lengths, from 10 to 20 seconds. Notably, the CLAP score for this experiment is measured using a different checkpoint from the one used in other tables, as experiments involving varying audio lengths require a distinct CLAP model[5].

Table 6: Comparison of generated audio lengths between a fixed duration of 10 seconds and variable-length generation approaches.

| Generated audio length | CLAP↑ | FD↓ | KL↓ |
|---|---|---|---|
| Fix | 0.3207 | **43.3788** | 1.9650 |
| Various | **0.3257** | 48.3701 | **1.9058** |

## 6 CONCLUSION

We introduce InfiniteAudio, a novel inference method designed to generate infinitely long, consistent audio using pretrained text-to-audio models. InfiniteAudio effectively maintains a fixed memory footprint, addressing the memory limitations of existing models. Additionally, we propose a new guidance alternation method that can produce long speech with high intelligibility. By sharing QKV pairs in the self-attention layers, InfiniteAudio ensures consistent speech generation and mitigates issues such as voice variations. These contributions open up possibilities for long text-to-audio generation and pave the way for continuous, coherent long audio content.

---

[5]https://github.com/LAION-AI/CLAP

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

## A  ALGORITHMS OF INFINITEAUDIO

We present pseudo-code for InfiniteAudio for TTA, TTLS, and TTAS respectively.

### A.1  TTA GENERATION WITH CURVED DENOISING

---

**Algorithm 1** InfiniteAudio for TTA

---

**Input**:

- $N$: number of frames
- $T$: audio frames
- $r$: compression factor
- $\frac{T}{r} = f$: total timesteps
- $\epsilon_\theta(\cdot)$: noise prediction model
- $\text{Dec}(\cdot)$: decoder
- $\{\mathbf{z}^i_{\tau_i}\}^f_{i=2}$: initial latent variables
- $\{\tau_i\}^f_{i=1}$: timesteps
- $\mathbf{c}_{desc}$: description prompt

**Output**: $v$: generated audio sequence

1: $v \leftarrow []$
2: $\boldsymbol{\tau} \leftarrow [\tau_1, \tau_{1+P}, \tau_{1+2P}; ...; \tau_{f-2}, \tau_{f-1}, \tau_f]$   ▷ Curved denoising with focused initial timesteps
3: $\boldsymbol{Q} \leftarrow [\mathbf{z}_1, \mathbf{z}_{1+P}, \mathbf{z}_{1+2P}; ...; \mathbf{z}_{f-2}, \mathbf{z}_{f-1}, \mathbf{z}_f]$   ▷ Latent variables for diffusion steps
4: **for** $l$ to $N$ **do**   ▷ Generate N frames of audio
5:     $\boldsymbol{Q} \leftarrow \epsilon_\theta(\boldsymbol{Q}, \boldsymbol{\tau}, \mathbf{c}_{desc}, \emptyset)$   ▷ Update latent variables with noise prediction
6:     $\mathbf{z}^l_{\tau_0} \leftarrow \boldsymbol{Q}.\text{dequeue}()$   ▷ Pop out the clean audio frame
7:     $v.\text{append}(\text{Dec}(\mathbf{z}_{\tau^l_0}))$   ▷ Decode the frame and add to output
8:     $\mathbf{z}^{l+\text{Len}(\boldsymbol{Q})}_{\tau_f} \sim \mathcal{N}(\mathbf{0}, \mathbf{I})$   ▷ Generate new random noise
9:     $\boldsymbol{Q}.\text{enqueue}(\mathbf{z}^{l+\text{Len}(\boldsymbol{Q})}_{\tau_f})$   ▷ Insert new noise to latent sequence
10: **end for**
11: **return** $v$   ▷ Return the generated audio sequence

---

## A.2 TTLS Generation with Guidance Alternation

---

**Algorithm 2** InfiniteAudio for TTLS (Text-to-Long Speech)

---

**Input**:

- $N$: Number of frames

- $T$: audio frames

- $r$: compression factor

- $f$: total timesteps, $\frac{T}{r} = f$

- $\boldsymbol{\epsilon}_{\theta_{odd}}(\cdot), \boldsymbol{\epsilon}_{\theta_{even}}(\cdot)$: Conditional/unconditional guidance models

- $\texttt{Dec}(\cdot)$: Decoder function to generate audio from latent states

- $\{\mathbf{z}_{\tau_i}^i\}_{i=2}^f$: Initial latent variables for each timestep

- $\{\tau_i\}_{i=1}^f$: Timestep schedule

- $\mathbf{c}_{desc}$: Description condition (high-level text description)

- $\mathbf{c}_{cont}$: Content conditions (detailed text split into segments)

- $sn$: Sentence number (starts at 0)

**Output**: Generated audio sequence $\boldsymbol{v}$

1: $\boldsymbol{v} \leftarrow []$  ▷ Initialize output audio
2: $\boldsymbol{\tau} \leftarrow [\tau_1, \tau_{1+P}, \tau_{1+2P}; ...; \tau_{f-2}, \tau_{f-1}, \tau_f]$  ▷ Curved denoising timestep schedule
3: $\boldsymbol{Q} \leftarrow [\mathbf{z}_1, \mathbf{z}_{1+P}, \mathbf{z}_{1+2P}; ...; \mathbf{z}_{f-2}, \mathbf{z}_{f-1}, \mathbf{z}_f]$  ▷ Initialize latent queue for diffusion
4: $[\mathbf{c}_{cont1}, \mathbf{c}_{cont2}, ..., \mathbf{c}_{contK}] \leftarrow \mathbf{c}_{cont}$  ▷ Split content prompts into segments
5: $S \leftarrow [\texttt{Len}(\mathbf{c}_{cont1}), \texttt{Len}(\mathbf{c}_{cont2}), ..., \texttt{Len}(\mathbf{c}_{contK})]$  ▷ Store segment lengths
6: $\boldsymbol{\epsilon}_\theta(\cdot) \leftarrow \boldsymbol{\epsilon}_{\theta_{odd}}(\cdot)$  ▷ Initialize with conditional guidance (odd)
7: **for** $l$ to $N$ **do**  ▷ Iterate over N frames
8:     **while** $l \leq S[sn]$ **do**  ▷ Generate frames for the current sentence
9:         $\boldsymbol{Q} \leftarrow \boldsymbol{\epsilon}_\theta(\boldsymbol{Q}, \boldsymbol{\tau}, \emptyset, \mathbf{c}_{cont})$  ▷ Apply noise prediction to latent states
10:         $\mathbf{z}_{\tau_0}^l \leftarrow \boldsymbol{Q}.\texttt{dequeue}()$  ▷ Pop out clean audio frame
11:         $\boldsymbol{v}.\texttt{append}(\texttt{Dec}(\mathbf{z}_{\tau_0}^l))$  ▷ Decode and append to output
12:         $\mathbf{z}_{\tau_f}^{l+\texttt{Len}(\boldsymbol{Q})} \sim \mathcal{N}(\mathbf{0}, \mathbf{I})$  ▷ Generate new random noise
13:         $\boldsymbol{Q}.\texttt{enqueue}(\mathbf{z}_{\tau_f}^{l+\texttt{Len}(\boldsymbol{Q})})$  ▷ Insert new noise into latent queue
14:         **if** $l = S[sn] - 1$ **then**  ▷ Check if we reached the end of the sentence
15:             $sn \leftarrow sn + 1$  ▷ Move to the next sentence
16:             **if** $\boldsymbol{\epsilon}_\theta(\cdot) = \boldsymbol{\epsilon}_{\theta_{odd}}(\cdot)$ **then**  ▷ Switch conditional guidance
17:                 $\boldsymbol{\epsilon}_\theta(\cdot) \leftarrow \boldsymbol{\epsilon}_{\theta_{even}}(\cdot)$  ▷ Switch to unconditional guidance
18:             **else**
19:                 $\boldsymbol{\epsilon}_\theta(\cdot) \leftarrow \boldsymbol{\epsilon}_{\theta_{odd}}(\cdot)$  ▷ Switch back to conditional guidance
20:             **end if**
21:         **end if**
22:     **end while**
23: **end for**
24: **return** $\boldsymbol{v}$  ▷ Return the final generated audio sequence

---

## A.3 INFINITEAUDIO FOR TTAS

---

**Algorithm 3** InfiniteAudio for TTAS (Text-to-Audio and Speech)

---

**Input**:

- $N$: Number of frames

- $T$: audio frames

- $r$: compression factor

- $f$: total timesteps, $\frac{T}{r} = f$

- $\boldsymbol{\epsilon}_{\theta_{odd}}(\cdot), \boldsymbol{\epsilon}_{\theta_{even}}(\cdot)$: Conditional/unconditional guidance models

- $\text{Dec}(\cdot)$: Decoder function to generate audio from latent states

- $\{\mathbf{z}_{\tau_i}^i\}_{i=2}^f$: Initial latent variables for each timestep

- $\{\tau_i\}_{i=1}^f$: Timestep schedule

- $\mathbf{c}_{desc}$: Description condition (high-level text description)

- $\mathbf{c}_{cont}$: Content conditions (detailed text split into segments)

- $sn$: Sentence number (starts at 0)

**Output**: Generated audio sequence $v$

1: $\boldsymbol{v} \leftarrow []$     ▷ Initialize output audio
2: $\boldsymbol{\tau} \leftarrow [\tau_1, \tau_{1+P}, \tau_{1+2P}; ...; \tau_{f-2}, \tau_{f-1}, \tau_f]$   ▷ Curved denoising timestep schedule
3: $\boldsymbol{Q} \leftarrow [\mathbf{z}_1, \mathbf{z}_{1+P}, \mathbf{z}_{1+2P}; ...; \mathbf{z}_{f-2}, \mathbf{z}_{f-1}, \mathbf{z}_f]$   ▷ Initialize latent queue for diffusion
4: $[\mathbf{c}_{cont1}, \mathbf{c}_{cont2}, ..., \mathbf{c}_{contK}] \leftarrow \mathbf{c}_{cont}$   ▷ Split content prompts into segments
5: $S \leftarrow [\text{Len}(\mathbf{c}_{cont1}), \text{Len}(\mathbf{c}_{cont2}), ..., \text{Len}(\mathbf{c}_{contK})]$   ▷ Store segment lengths
6: $\boldsymbol{\epsilon}_{\theta}(\cdot) \leftarrow \boldsymbol{\epsilon}_{\theta_{odd}}(\cdot)$   ▷ Initialize with conditional guidance (odd)
7: **for** $l$ to $N$ **do**     ▷ Iterate over N frames
8:   **while** $l \leq S[sn]$ **do**   ▷ Generate frames for the current sentence
9:    $\boldsymbol{Q} \leftarrow \boldsymbol{\epsilon}_{\theta}(\boldsymbol{Q}, \boldsymbol{\tau}, \mathbf{c}_{desc}, \mathbf{c}_{cont})$   ▷ Apply noise prediction to latent states
10:    $\mathbf{z}_{\tau_0}^l \leftarrow \boldsymbol{Q}.\text{dequeue}()$   ▷ Pop out clean audio frame
11:    $\boldsymbol{v}.\text{append}(\text{Dec}(\mathbf{z}_{\tau_0}^l))$   ▷ Decode and append to output
12:    $\mathbf{z}_{\tau_f}^{l+\text{Len}(\boldsymbol{Q})} \sim \mathcal{N}(\mathbf{0}, \mathbf{I})$   ▷ Generate new random noise
13:    $\boldsymbol{Q}.\text{enqueue}(\mathbf{z}_{\tau_f}^{l+\text{Len}(\boldsymbol{Q})})$   ▷ Insert new noise into latent queue
14:    **if** $l = S[sn] - 1$ **then**   ▷ Check if we reached the end of the sentence
15:     $sn \leftarrow sn + 1$   ▷ Move to the next sentence
16:     **if** $\boldsymbol{\epsilon}_{\theta}(\cdot) = \boldsymbol{\epsilon}_{\theta_{odd}}(\cdot)$ **then**   ▷ Switch conditional guidance
17:      $\boldsymbol{\epsilon}_{\theta}(\cdot) \leftarrow \boldsymbol{\epsilon}_{\theta_{even}}(\cdot)$   ▷ Switch to unconditional guidance
18:     **else**
19:      $\boldsymbol{\epsilon}_{\theta}(\cdot) \leftarrow \boldsymbol{\epsilon}_{\theta_{odd}}(\cdot)$   ▷ Switch back to conditional guidance
20:     **end if**
21:    **end if**
22:   **end while**
23: **end for**
24: **return** $v$     ▷ Return the final generated audio sequence

---

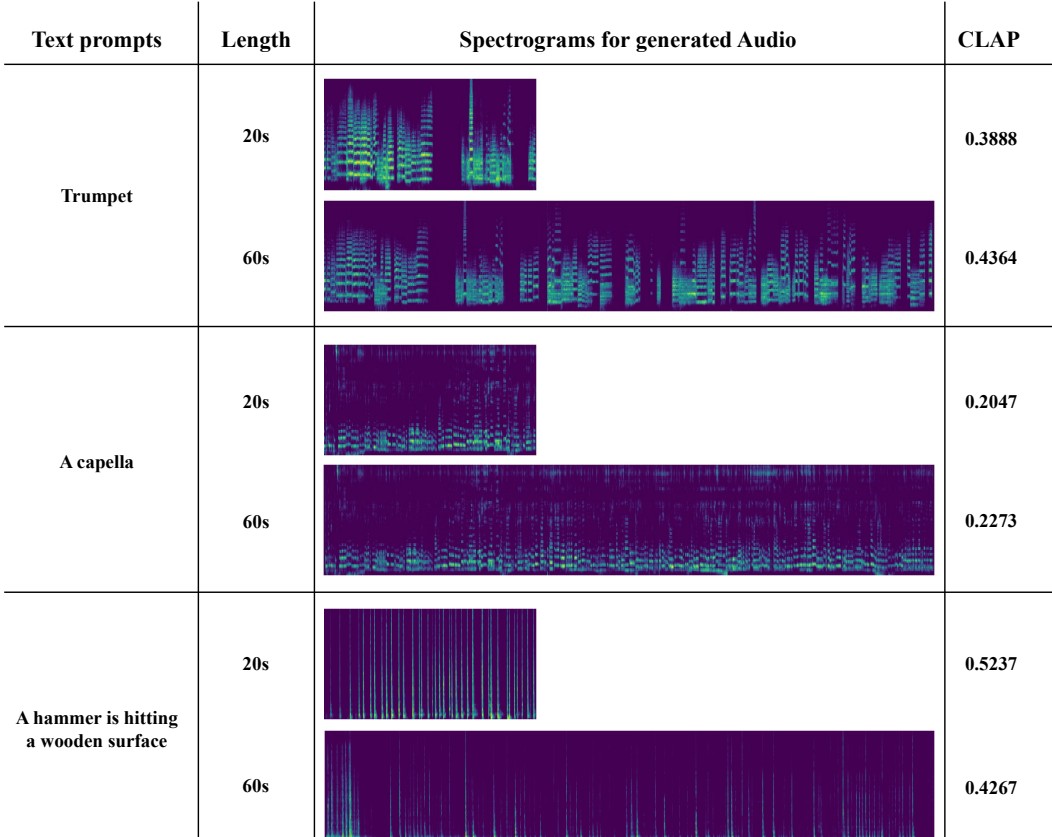

| Text prompts | Length | Spectrograms for generated Audio | CLAP |
|---|---|---|---|
| Trumpet | 20s | | 0.3888 |
| | 60s | | 0.4364 |
| A capella | 20s | | 0.2047 |
| | 60s | | 0.2273 |
| A hammer is hitting a wooden surface | 20s | | 0.5237 |
| | 60s | | 0.4267 |

Figure 10: Generated audio samples based on AudioLDM.

## B  GENERATED SAMPLES

### B.1  AUDIOLDM

We present additional audio samples that showcase the capabilities of InfiniteAudio, which builds upon AudioLDM. As illustrated in Fig. 10, InfiniteAudio can generate a variety of sounds exceeding 10 seconds in duration. Remarkably, the generated audio maintains consistency with the provided text prompts for up to 20 seconds. Furthermore, the system demonstrates the ability to produce coherent audio for durations of up to 60 seconds, as evidenced by CLAP scores.

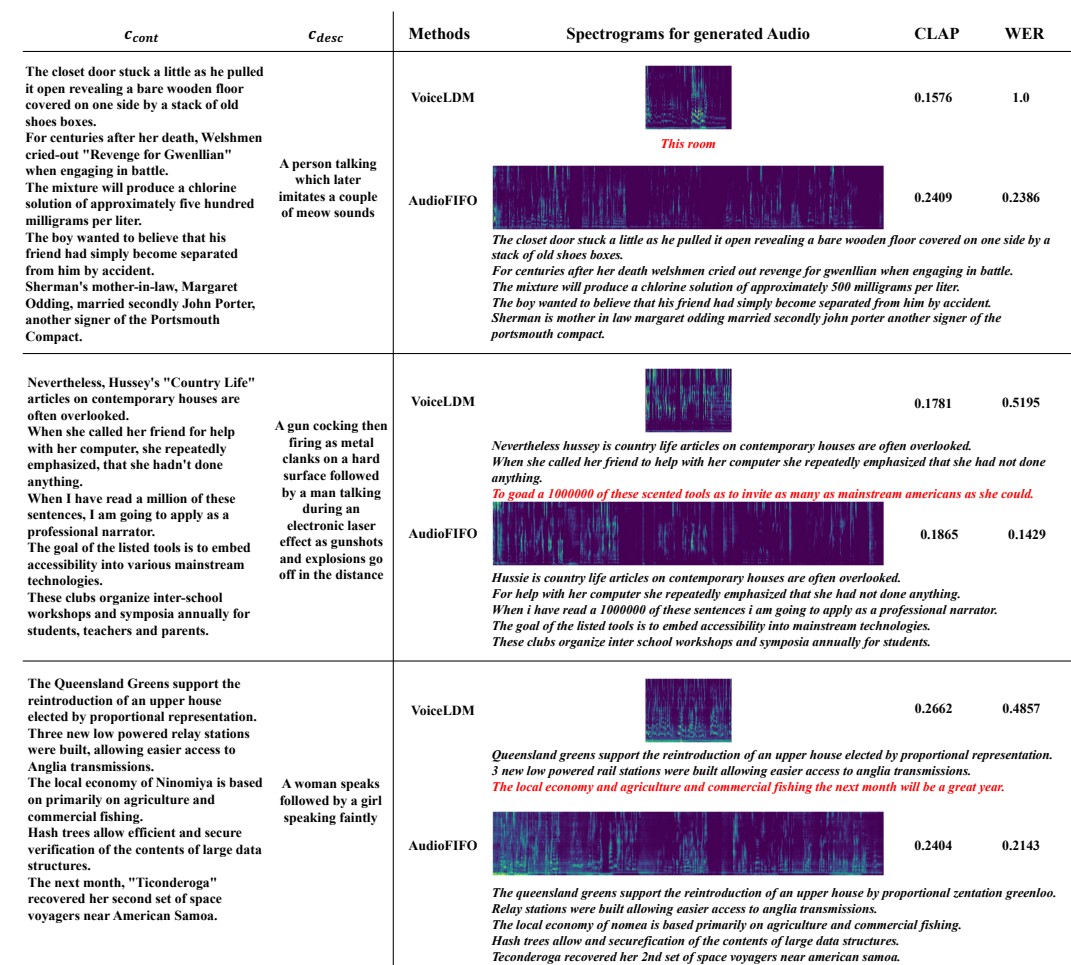

Figure 11: Generated audio samples based on VoiceLDM.

## B.2 VOICELDM

In our exploration of VoiceLDM, we generate audio based on two distinct text prompts: $c_{cont}$ and $c_{desc}$. As shown in Fig. 11, our approach produces significantly improved audio quality when provided with longer content prompts. However, it's worth noting that VoiceLDM is limited to generating audio of no more than 10 seconds, often resulting in truncated or incomplete sentences. For clarity, we have included the transcriptions of the sentences below each spectrogram, with any cut-off portions highlighted in red. In contrast, our method ensures the generation of entire sentences without omitting sections. The superiority of our approach is further supported by CLAP and WER scores, which validate the enhanced intelligibility and coherence of the generated audio.

## C  EXPERIMENT DETAILS

### C.1  DATASET

**Existing TTA models:**  AudioLDM is trained on a diverse combination of datasets, including AudioSet (Gemmeke et al., 2017), the largest audio dataset with over 5,000 hours of data, as well as AudioCaps Kim et al. (2019), Freesound (FS), and the BBC Sound Effect (SFX) library, covering a wide range of sounds. Similarly, VoiceLDM is trained on AudioSet for TTA, the English subset of the CommonVoice 13.0 corpus and VoxCeleb1 for speech generation, and the DEMAND dataset for non-speech segments. AudioLDM is evaluated on both the AudioSet and AudioCaps datasets, while VoiceLDM is tested exclusively on the AudioCaps dataset.

**InfiniteAudio:**  We utilize the Audiocaps test set for text-to-audio generation, which comprises audio files paired with corresponding caption texts. Each audio file is accompanied by several text captions, from which we randomly select 860 audio-text pairs for evaluation.

For text-to-speech (TTS) evaluation, we employ the CommonVoice 13.0K test set. Unlike traditional TTS evaluations, our focus is on generating longer speech segments. Therefore, we specifically target sentences exceeding 90 characters in length. For each text input, we utilize more than four selected sentences, resulting in a total of 60 text pairs and approximately 300 sentences for testing.

Both audio and speech generation evaluations leverage the aforementioned datasets. We include captions categorized as "speech" from the Audiocaps test set as prompts for audio descriptions and randomly select 60 long speech pairs from the CommonVoice 13.0K test set for content prompts.

### C.2  CONFIGURATION

We conducted experiments using InfiniteAudio alongside existing text-to-audio generation models, AudioLDM and VoiceLDM. Both models are based on Latent Diffusion Models (LDM) utilizing a U-Net architecture. We increased the number of inference steps to optimize performance, deviating from the default settings of the original models, and employed DDIM sampling.

For AudioLDM, we set the inference steps to 300, aligning with the original model but omitting the initial and middle regions, except for multiples of 4, while retaining the final steps, which introduce slight noise into the spectrograms. In contrast, for VoiceLDM, we increased the inference steps from the original 50 to 200, skipping the middle and final regions except for multiples of 5, while including the initial timesteps to introduce Gaussian-like noise.

In text-to-long speech generation (TTLS), we segment long content prompts at the sentence level. For QKV sharing, we apply the sharing mechanism every 200 audio frames. However, at the start of each new sentence, the sharing process is reset, beginning again with the first 200 frames.

### C.3  HUMAN EVALUATION

Subjective evaluation plays a vital role in the text-to-audio generation domain. For our assessment, we randomly selected 30 generated audio samples, which were rated by 20 domain experts on a scale from 1 to 5. The evaluation criteria focused on overall audio quality and the relevance of the generated audio to the corresponding descriptive text.

Our model aims to generate longer audio segments while preserving the performance of pretrained text-to-audio models. Despite the inherent challenges in evaluating these samples, our results indicate comparable performance to existing models, with ground truth scores averaging around 4.

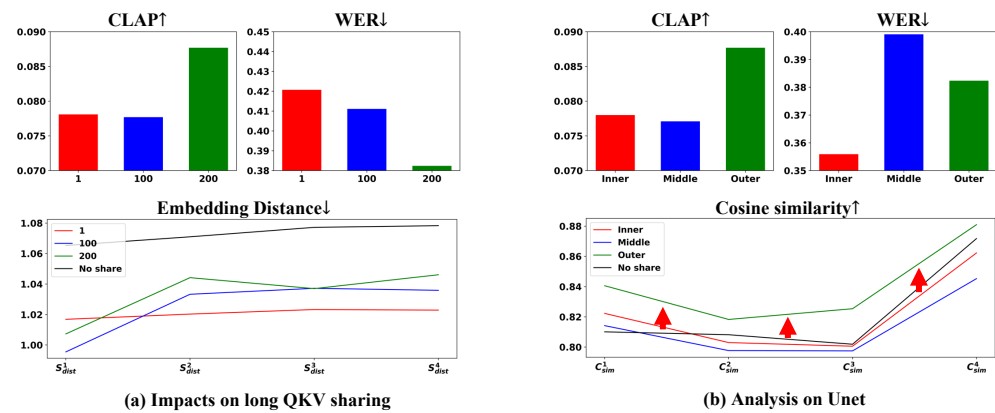

(a) Impacts on long QKV sharing       (b) Analysis on Unet

Figure 12: Analysis of QKV sharing: (a) Impact of extended QKV sharing and (b) Comparison of U-Net decoder modules.

# D ADDITIONAL EXPERIMENTS

## D.1 IMPACT OF EXTENDED QKV SHARING

As illustrated in Fig. 12, sharing Query, Key, and Value (QKV) representations over 200 audio frames proves effective based on CLAP, Word Error Rate (WER), and speaker embedding distance scores. While sharing QKV for a single frame yields smaller speaker embedding distances, it can adversely affect CLAP and WER metrics. In contrast, sharing QKV across 200 frames reduces speaker embedding distances compared to the case with no sharing, while simultaneously enhancing both CLAP and WER scores.

## D.2 COMPARATIVE ANALYSIS OF U-NET DECODER MODULES

The U-Net architecture comprises both encoder and decoder components, typically organized into multiple downsampling and upsampling modules. In this study, we categorize these modules into inner, middle, and outer groups. Previous works, such as (Ceylan et al., 2023; Kwon et al., 2024; Tumanyan et al., 2023), have explored sharing mechanisms across various U-Net decoder modules. We empirically determined the most effective module for sharing. Our findings indicate that the outer module enhances speaker embedding similarity and improves CLAP scores.

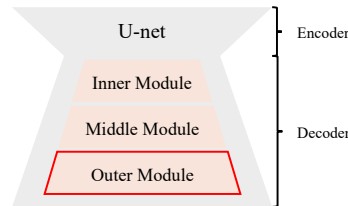

Figure 13: Architecture of the U-Net decoder.

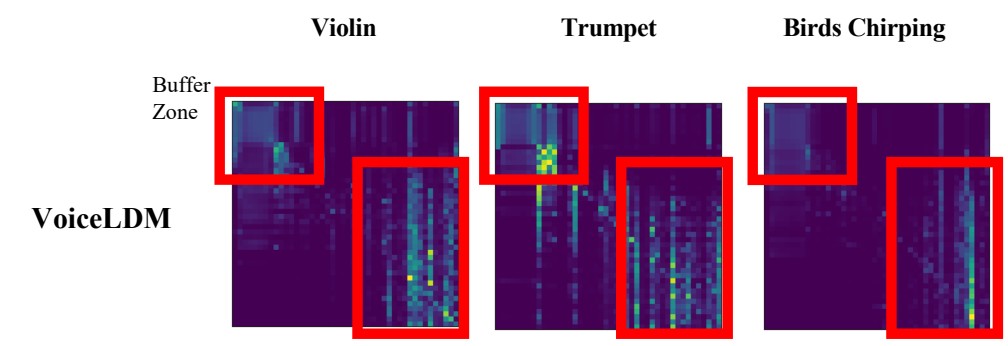

Figure 14: Attention map analysis for robustness of sampling strategies.

## D.3 ATTENTION MAP ROBUSTNESS

As shown in Fig. 14, VoiceLDM emphasizes the initial diffusion timesteps corresponding to the later parts of the key matrices as the query sequences approach their endpoints. Moreover, the attention maps are similar across different description prompts. This indicates that the sampling strategy focusing on the initial timesteps is robust for all samples.

### D.4 EXISTING LONG GENERATION METHODS BASED ON DIFFUSION MODELS.

We briefly review key papers on long-form generation using pretrained diffusion models, focusing specifically on methods that do not involve additional training in other generative domains.

#### D.4.1 MULTIDIFFUSION FOR TTI

MultiDiffusion[6] is a robust framework for text-to-image generation that leverages a pre-trained diffusion model without requiring additional training or fine-tuning. At its core, MultiDiffusion merges multiple independent diffusion processes through an optimization algorithm, reconciling them into a coherent and high-quality image. This approach enables user-controllable image generation, making it highly adaptable for a wide range of tasks.

The primary innovation of MultiDiffusion lies in its ability to simultaneously process multiple image regions, adhering to user-defined constraints such as text prompts, aspect ratios, and spatial layout signals (e.g., segmentation masks or bounding boxes). However, a key limitation of the framework is its inability to incorporate temporal information, restricting its utility to image generation. This makes it unsuitable for video generation, where maintaining temporal coherence across frames is essential.

Despite this limitation, the underlying optimization process ensures that all image regions conform closely to the reference diffusion model, preserving both high image quality and visual consistency. While not applicable to temporal tasks, MultiDiffusion remains a flexible and efficient solution for generating complex images that meet diverse spatial constraints.

#### D.4.2 FREENOISE FOR TTV

FreeNoise[7] is a framework designed to extend the capabilities of text-to-video diffusion models for generating longer, temporally coherent videos. Traditional text-to-video models are typically trained on a limited number of frames, restricting their ability to generate high-fidelity long videos during inference. FreeNoise addresses this limitation by introducing a **tuning-free** paradigm that dynamically reschedules noise over time to maintain consistency across frames.

**Key Innovations**

- **Noise Rescheduling**: Unlike traditional methods that initialize noise uniformly for all frames, FreeNoise dynamically adjusts the noise distribution during video generation. This method captures long-range temporal correlations, ensuring that visual consistency is preserved across extended video sequences.

- **Temporal Attention Mechanism**: FreeNoise incorporates a window-based temporal attention mechanism, which helps maintain coherence over longer time frames. By focusing attention over localized windows, the model can efficiently capture and retain relevant temporal dependencies.

- **Motion Injection for Multi-Prompt Videos**: The framework supports multi-prompt video generation by enabling dynamic changes in video content based on evolving text prompts. This allows FreeNoise to generate videos where different segments adhere to different prompts, accommodating more complex narrative transitions over time.

**Limitations** Despite its strengths, FreeNoise has certain limitations. Since the framework does not involve any fine-tuning of the pre-trained models, it might not be optimally adapted to domain-specific datasets, potentially leading to suboptimal performance in specialized contexts. Moreover, while it addresses temporal consistency, the model's reliance on pre-trained diffusion models can limit its ability to handle diverse or complex motion dynamics inherent in specific generative tasks.

---

[6]https://github.com/omerbt/MultiDiffusion
[7]https://github.com/AILab-CVC/FreeNoise

