# OpenReview forum: "InfiniteAudio: Infinite-Length Audio Generation with Consistent Acoustic Attributes"
_ICLR.cc/2025/Conference — ICLR 2025 Conference Withdrawn Submission_

### Official Review · Reviewer_dwje · 2024-10-28

**Soundness:** 3
**Presentation:** 2
**Contribution:** 3
**Rating:** 3
**Confidence:** 4

**Summary:**

InfiniteAudio presents a novel method for generating long-duration audio sequences that maintain acoustic coherence. It is based on diffusion models by implementing a curved denoising approach with a fixed-size input, introducing conditional guidance alternation, and sharing initial self-attention features across future frames.

**Strengths:**

1. Infinite Audio Generation: The paper introduces a method to generate audio of unlimited length, which is a significant advancement over existing models that are limited by memory constraints.
2. Memory Efficiency: InfiniteAudio's curved denoising approach maintains a fixed memory footprint regardless of the output length, making it highly efficient for resource-limited environments.
3. Selective Sampling Steps: The paper's approach to selective denoising, focusing on the most important steps, reduces the number of required sampling steps while maintaining high-quality generation, leading to more efficient inference.

**Weaknesses:**

1. Insufficient innovation. This paper is more like solving an engineering problem rather than a scientific research problem. The methods used are mostly tricks rather than general methods. For example, if we switch from AudioLDM to Tango/Stable Audio, is the proposed method applicable? Do the parameters need to be further adjusted? This method regards mel-spectrogram as a picture, and borrows methods more from vision, without analyzing the characteristics of the audio itself.
2. The dataset and experimental results of baselines are deviated from the popular SOTA methods. Why does TTA only use 500 audio samples from audiocaps as the testset instead of all of them? At present, the FD of the classic method of audio generation can reach about 20. The FD of the AudioLDM in Table 2 is much worse than the result reported in the original paper. What is the reason?
3. Lack of experimental details: The paper aims to generate long audio. How long is the audio generated in the experiments? An important innovation of the paper is the adjustment of the sampling step length. What is the specific length? For example, what is the parameter P in the appendix A?
4、The main problem is that the generated audio is poor. I listen to some demos and the generated speech has the problem of fluctuating sound energy. This problem is mainly due to the lack of consistency and temporal coherence between the audio segments. The paper does not discuss and test this. The generated long-duration audio has noise and distortion. In the later stage of the generation, the generation collapses. I suspect that this is due to the sampling step size problem. The paper lacks a discussion on this issue.

**Questions:**

1. In Sec 5.2, memory usage depends on the specific machine. In the paper, you should introduce the GPU you used.
2. Some questions about the experimental results. In Table 2, are the results reported by AudioLDM and VoiceLDM based on the 10s-generated audio? Or other lengths? How long is the audio generated by InfiniteAudio in different comparison settings? Similarly, in Table 3, does voiceLDM-base generate all speech in 10s? Is this unreasonable in itself? Why is the WER so poor?
3. Have you tried to measure the consistency and temporal coherence of audio between different segments? Resemblyzer and VoxCeleb-disentangler only evaluate the speaker similarity but did not compare the consistency of audio content between segments.
4. One of the simplest ways to generate long-term audio is to generate multiple small segments of audio and then splice them together. Can you compare the proposed method with this simple chunk-wise generation?

---

### Official Review · Reviewer_a5k7 · 2024-11-01

**Soundness:** 2
**Presentation:** 1
**Contribution:** 2
**Rating:** 3
**Confidence:** 4

**Summary:**

The paper proposes a system called InfiniteAudio to generate audio, speech or audio-speech simultaneously, theoretically supporting infinite duration.
InfiniteAudio could be interpreted as a set of inference-time techniques, thus involves no additional training or finetuning, and is compatible with existing & off-the-shelf pretrained diffusion models.
InfiniteAudio tackles the challenges in long-form audio or speech generation, including
1. Memory usage
2. Computational cost
3. Performance drop
4. Inconsistency along the time axis

With techniques such as
1. An inference strategy inspired by FIFO (first-in, first-out) that fixes the size of the noisy latents for the diffusion model.
2. Reducing diffusion timesteps by skipping unimportant timesteps
3. Guidance alternation
4. Shared self-attention features (QKV embeddings)

However, several issues relating to writing and evaluation are observed, which in my opinion affected the quality of this paper.

**Strengths:**

- The challenges for long-form audio/speech-audio generation listed in this paper are solid.
- The improvement to VoiceLDM is solid.

**Weaknesses:**

## Major concerns
### 1. improper evaluation metrics
Table.3 uses the word error rate (WER) as a metric for the evaluation of audio-speech simultaneous generation (TTAS), and interprets the lower WER scores as "better" result.

The definition of TTAS follows that of VoiceLDM: synthesizing speeches according to not only the content of speech, but also a "descriptive prompt" that specifies gloabl acoustic features.

According to the task definition, when the descriptive promt indicates a noisy environment, "speeches accompanied with very noisy environmental sound" could be the correct generation result, and such a noisy speech may lead to higher WER score. Consequently, a higher WER score in the TTAS part of table.3 does not necessarily indicates a worse generation.

A support to the above observation is that,  the CLAP score of the proposed method is worse than all other 3 methods, while its WER is better than others. If the model less reflects "noisy descriptive prompts", WER can be improved, but we cannot conclude such a behavior as a good one.
### 2. Less contribution to TTS community
As mentioned in line 424 & 425 in the manuscript, the high WER of VoiceLDM (baseline for TTLS and TTAS) is partly because VoiceLDM cannot include all speech contents within 10 seconds.
InfiniteAudio further improves the "speaker inconsistency" problem in VoiceLDM. Although there is no explicit explanation on the cause of this inconsistency, I assume it is because of the lack of an explicit speaker embedding in VoiceLDM.

After listening to the demo page, I totally agree that InfiniteAudio greatly improved the above issues of VoiceLDM, but I don't think this improvement is reusable to the broad community, as these improvements are quite limited to the design of VoiceLDM, a model that is very different from modern TTS systems.
1. Bad WER caused by fixed duration generation. Modern TTS systems does not assume a fixed duration of the output, and can predict a proper duration or speech style according to the content text. Go to the demo page of [styletts2], the first sample is 1 min 29s long. Modern TTS systems will not miss content text in their generated results.
2. Bad speaker consistency caused by the lack of explicit speaker embedding. InfiniteAudio address this problem by sharing the identica QKV features across the time axis. I think the QKV features implicitly contain the speaker id. So the proposed "sharing QKV" technique is conceptually similar to "introducing a speaker embedding to VoiceLDM", which is the common practice in modern TTS systems.

Although I understand VoiceLDM is the most available TTAS model by now, earlier works such as [VALL-E] can already generate noisy speeches when a noisy audio prompt is given, so it is unclear to me which one is more valuable, keep improving VoiceLDM or introduce the support of descriptive prompt to successful TTS models. Solving problems that do not exist in other (more common) TTS methods hardly contribute to the broader community.

[styletts2]: https://styletts2.github.io/
[VALL-E]: https://arxiv.org/abs/2301.02111
## Minor concerns
### 1. unclear definition in paper writing.
After reading Sec.4, I thought the term "InfiniteAudio" means the combination of techniques listed in all sub-sections of Sec.4.
However, in table.2, table.3 and table.4, we can frequently observe terms like "InfiniteAudio w/ xxx technique". This indicates an inconsistent definition of the term "InfiniteAudio" across different tables, which is confusing.
### 2. unclear testing condition for baselines and proposed models
"InfiniteAudio" can generate audio of arbitrary duration, hence it is important to clarify the targeted duration within an evaluation.

Unfortunately, the targeted duration is missing for table.2. In the demo page, durations of 10s, 20s, 30s, 1min and 2min are included. If scores in table.2 are computed with mixed durations, I doubt if these scores are meaningful. Mixed duration can significantly affect subjective listening tests.
### 3. Missing baselines that can be simply implemented
The FIFO (first-in, first-out) strategy is interesting, but the concatenation-based strategy mentioned in line 46 is never evaluated in the paper. I think it is important to compare the FIFO strategy with an overlap-concatenate stategy, as both of them can control memory usage by fixing the input size of the diffusion model.

%%%It is know that AudioLDM can do time inpainting, i.e., given the first 4-sec and some descriptive text, AudioLDM can naturally inpaint the remaining 6-sec. If we combine this inpainting capability with concatenation, we can generately very long audio with a fixed input size.

**Questions:**

- What is the targeted duration in table.2?
- In table.2, the "InfiniteAudio w/ Last focused timesteps" is obviously worse than the baseline AudioLDM in terms of subjective metrics, could you explain more about this?
- In figure.9 cosine similarity figure, we can see the blue curve (w/o Q) is more advantages for $C^2_{sim}, C^3_{sim}, C^4_{sim}$, why do the authors claim "sharing QKV features consistently outperforms other approaches across all metrics" in line 484?

---

### Official Review · Reviewer_AJ4H · 2024-11-02

**Soundness:** 2
**Presentation:** 2
**Contribution:** 2
**Rating:** 3
**Confidence:** 2

**Summary:**

The paper addresses the challenges on generating longer audio sequences, by introducing curved denoising. It also introduces a conditional guidance alternation mechanism to maintain intelligibility of  generated long speech. To ensure consistent speech generation, QKV sharing is also introduced.

**Strengths:**

The paper proposes a interesing method to generate longer audio data maintaining constant memory usage.

**Weaknesses:**

- Theoretical analysis of proposed method is not shown enough. Especially, theoretical validity of using multiple diffusion steps at the same time.
- Limited applicability. It  proposes a way for addressing the issues of VoiceLDM, but not clear if it is also beneficial for other diffusion models and task setting.

**Questions:**

- In 4.2, performance should depend on how the long sentence token is divided into several sentence tokens. Have you tested different ways of how it is divided.
- Have you confirmed theoritical validity of using both of conditional and unconditional guidance? Is it empirically derived?
- Are the methods shown in 4.2 and 4.3 effective only for TTLS and TTAS task rather than text-to-audio (audio without speech)?
- In the experiments that WER are used, have you tested it with speech recognition model other than Whisper. The results are heavily affected by characteristics of Whisper
- In Table 4, Why did you adopt Resemblyzer in the experiment? have you tested speaker embeddings extractor other than Resemblyzer?

---

### Official Review · Reviewer_ptTT · 2024-11-04

**Soundness:** 3
**Presentation:** 2
**Contribution:** 3
**Rating:** 6
**Confidence:** 3

**Summary:**

This work proposes a training-free TTA/TTAS algorithm which uses pretrained diffusion models to generate audio in arbitrary length while still having a constant-size memory footprint. It includes several techniques such as splitting the content prompt for long speech content, guidance alternation o improve intelligibility, optimized timestep scheduling(i.e. curved denoising) and QKV sharing which helps preserving speaker identity. The proposed algorithm reached comparable audio quality against baselines(AudioLDM, VoiceLDM) with optimized timesteps, and its speech intelligibility surpassed the VoiceLDM baseline. Several ablations included in the paper aimed to verify the effectiveness of the proposed techniques.

**Strengths:**

In general, the proposed algorithm has several attractive properties, such as training-free, capability of arbitrary length generation, TTAS capability rather than only TTA or TTS. It proposed several interesting techniques to solve issues encountered in using diffusion model for arbitrary length generation. Some of the techniques even have potential to be used in other places such as guidance alternation and QKV sharing. The audio quality of the sample on the demo site is acceptable and perceptually align with the metrics reported in the paper.

**Weaknesses:**

Overall, the major weakness of this paper is about the clarity. Details of techniques are not well disclosed and the some of the terms in equations are not explained. Another concern is about the generalizability of the timestep optimization. If this is dataset dependent and only a very specific scheduling can yield good result, it'll become an obstacle for people to apply this method in future works.

Please find detailed comments on clarity below:

- The explanation about QKV sharing is insufficient. I would suggest to show the formulas explicitly, or add reference to previous works. I suppose here the proposed work adapted the feature injection in [Tumanyan et al., 2023]. Considering the proposed model shifts 1 frame per timestep, sharing QKV at the first N frame is equivalent to sharing QKV in the first N steps of the backward diffusion, which is a reasonable setup. However, in Appendix D1, there's a case of "QKV sharing on 1 frame". If there's only one frame, it should be the same as "no sharing"?

[Tumanyan et al., 2023] Narek Tumanyan, Michal Geyer, Shai Bagon, and Tali Dekel. Plug-and-play diffusion features for text-driven image-to-image translation. In Proc. CVPR, 2023.

- Regarding the guidance alternation, what's the definition of $\epsilon^{odd}$ and $\epsilon^{even}$? From the text context I could infer that the idea is to remove the effect of $c_{cont}$ from the previous segment, but this is not shown in Eq 4 and 5, in which all the $c_{cont}$ are without indices.

- Regarding the timestep allocation about the curved denoising, the definition of "initial", "middle" and "final" region should be stated clearly. Another concern is,  it could be that each dataset requires different schedule. This could make it difficult to apply this strategy in other applications. Moreover, based on those figures in Appendix D3, although the attention map share some common pattern for different prompts, the entire map still varies across prompts. This could mean that the proper partitioning of initial, middle and final region is sample dependent.

- Although with guidance alternation and QKV sharing, the CLAP score of proposed method is still lower than the VoiceLDM baseline on TTAS. Furthermore, adding QKV sharing seems impacted the CLAP score. This could be a hard trade-off on whether to use QKV sharing in practical applications.

- Regarding the claim about the quality of proposed method, in figure 8, it's claimed that proposed work generates "high-quality" long audio. But when I listened to the sample with same prompt on the demo site, I can clearly perceive the artifact. The intelligibility of the sample is acceptable, just that the voice timbre and some background sound are clearly distorted. Moreover, when compared to VoiceLDM in TTAS, the voice quality of VoiceLDM is still clearly better than proposed work. On the other hand, the background sound of TTAS is distorted (e.g. motorcycle engine) for both models (AudioLDM and InfiniteAudio).

**Questions:**

- What's the actual difference between $\epsilon$, $\epsilon^{odd}$ and $\epsilon^{even}$? In the appendix they're called as "conditional" and "unconditional" guidance. But in Eq 4 and 5 they're both parametrized in conditional and unconditional form.

- In section 5.1 : why not simply use all the AudioCaps test set here?

- Is the 60 randomly selected test set used in TTAS evaluation the same as the one used in TTLS evaluation?

- In speaker(voice) consistency eval(table 4), Why it is settled as first 10s vs subsequent 5s sequences? Since the proposed method can generate arbitrary length, having 10s for all subsequent sequences is more reasonable in my opinion.

- In section 5.3 : How long is the sample you shown in Fig. 7? Is the CLAP score evaluated against the ground truth?

- In section 5.5, L527: Why not always use the CLAP checkpoint with feature fusion? If this is possible, then there's no need to switch between the checkpoints.

- In section 5.5, regarding the comparison of sampling steps between VoiceLDM and proposed method, is timestep scheduling the only difference here?

---

### Note · Authors · 2024-11-18

I have read and agree with the venue's withdrawal policy on behalf of myself and my co-authors.